# EFFICIENT FACIAL LANDMARK DETECTION VIA PRIOR KNOWLEDGE-GUIDED AGENTS

## ABSTRACT

We present a highly efficient, agent-based framework for facial landmark detection that prioritizes model compactness and computational efficiency over maximum accuracy. Unlike conventional approaches that rely on large, fully supervised models, our method assigns each agent to a specific landmark, enabling it to infer its position solely from local observations and prior knowledge without explicit location awareness or inter-agent communication. Prior knowledge is modeled in two embedding spaces—feature and coordinate—using class-conditional Gaussian distributions. Agents navigate by minimizing deviations from these priors via a lightweight policy network. To enhance representation learning, we introduce a proximity-weighted contrastive learning strategy that incorporates spatial proximity into the training objective. A multi-stage detection strategy further reduces redundant computation by detecting sub-landmarks relative to core landmarks. While our method produces slightly higher normalized mean error than state-of-the-art (SoTA) methods, it achieves over $16\times$ and $41\times$ improvements in space and time complexities, respectively, compared to the SoTA lightweight model, running at $4.19$ and $1.29$ frames per second on an i5 CPU ($2.5$ GHz) for the COFW and 300W datasets, respectively.

## 1 INTRODUCTION

Facial landmark detection is a fundamental component in many computer vision applications, including face recognition (Zhao et al., 2003), expression analysis (Yang et al., 2018), and 3D face reconstruction (Liu et al., 2018). Over the past decade, advances in deep learning have greatly improved detection accuracy. However, the increasing demand for real-time performance on resource-constrained platforms, such as mobile devices, AR/VR headsets, and embedded AI modules, has shifted attention toward efficiency-oriented solutions. Such platforms impose strict limits on power consumption, computation, and memory, motivating the need for new algorithms that balance accuracy with efficiency.

Facial landmark detection aims to localize key facial points in 2D images. State-of-the-art (SoTA) methods often achieve high accuracy using large-scale models, especially those based on supervised heatmap and coordinate regression (Feng et al., 2018; Lin et al., 2021; Wang et al., 2020; Huang et al., 2021; Dang et al., 2025). However, these models incur high computational and memory costs, making them unsuitable for embedded or low-power environments. In contrast, our method adopts a lightweight, agent-based approach that leverages prior knowledge and local observations to detect landmarks efficiently. Although our normalized mean error (NME) is higher than that of SoTA models, our approach offers orders-of-magnitude gains in efficiency with only 577k parameters and $< 30$ MFLOPs, compared to 9.7–67M parameters and 1.2–26.8 GFLOPs for conventional methods. This trade-off makes our method highly practical for embedded applications.

Our contributions are as follows:

- We propose an agent-based framework in which each agent independently localizes a specific landmark using only local observations and learned priors without access to absolute coordinates.
- We model prior knowledge in both latent feature and coordinate spaces via class-conditional generative models, enabling effective search under weak supervision.

- We introduce a spatially aware contrastive learning method that weights positives according to spatial proximity, improving robustness in occluded or noisy conditions.

- Despite slightly higher NME than SoTA models, our method achieves $16.8\times$ lower space complexity and $41.1\times$ lower time complexity than the best lightweight baseline, enabling CPU inference.

## 2 RELATED WORK

### 2.1 REGRESSION-BASED METHODS

Facial landmark detection methods are commonly categorized into coordinate regression (CR) and heatmap regression (HR) approaches. CR methods (Feng et al., 2018; Qian et al., 2019; Lin et al., 2021) directly predict landmark coordinates using deep neural networks, learning both spatial mappings and local features. While conceptually straightforward, CR approaches are highly sensitive to noise and bias, and typically require extensive supervision to achieve accurate predictions. HR methods (Wang et al., 2020; Huang et al., 2021) generate heatmaps for each landmark, from which coordinates are derived via a decoding step. Despite their strong accuracy, HR models face two notable limitations:
**Quantization error**: Because heatmaps are typically of lower resolution than the input image, the decoding process introduces quantization errors (Bulat et al., 2021; Lan et al., 2021), degrading coordinate precision.
**Lack of landmark correlation modeling**: Standard HR methods generate heatmaps independently for each landmark, ignoring spatial relationships.
D-ViT (Dang et al., 2025) addresses this via spatial-split and channel-split vision transformers. Both CR and HR methods require full-image access and combine feature extraction with coordinate prediction in a single large model, leading to high parameter counts and computational cost. Mediapipe (Lugaresi et al., 2019) employs a extre,e lightweight regression pipeline with model-based refinement, enabling significantly fewer parameters and computational cost.

### 2.2 MULTIPLE LANDMARK DETECTION WITH AGENTS

Detecting multiple landmarks with agents is challenging due to the need for coordination and reliance on partial visual observations. A key difficulty lies in effectively leveraging prior knowledge of both morphological and spatial correlations among landmarks. MARL (Vlontzos et al., 2019) uses a Deep Q-Network with inter-agent communication to implicitly capture morphological relationships through joint actions. The Multiscale Agent (Alansary et al., 2019) method addresses spatial relations by incorporating multiscale search. SGMaRL (Wan et al., 2023) integrates a statistical shape model (Cootes et al., 1995) to refine landmark positions based on spatial structure. While these approaches consider landmark correlations, they do so only partially—handling either morphological or spatial aspects in isolation. None fully integrate both dimensions of prior knowledge, limiting their ability to guide agents efficiently and accurately in complex visual conditions.

### 2.3 CONTRASTIVE LEARNING

Contrastive learning projects representations into a latent space where similar instances are pulled together and dissimilar ones are pushed apart (Chen et al., 2020; Tian et al., 2020). Typically, two augmented views per sample yield $2N$ views for a batch of $N$ samples. In self-supervised settings, only views from the same source are considered positive pairs. A limitation is that different-class instances may be treated as negatives, even if semantically related. Supervised Contrastive Loss (SupConLoss) (Khosla et al., 2020) addresses this by incorporating label information, allowing multiple positive pairs per class. This improves representation quality, leading to more accurate and robust classification.

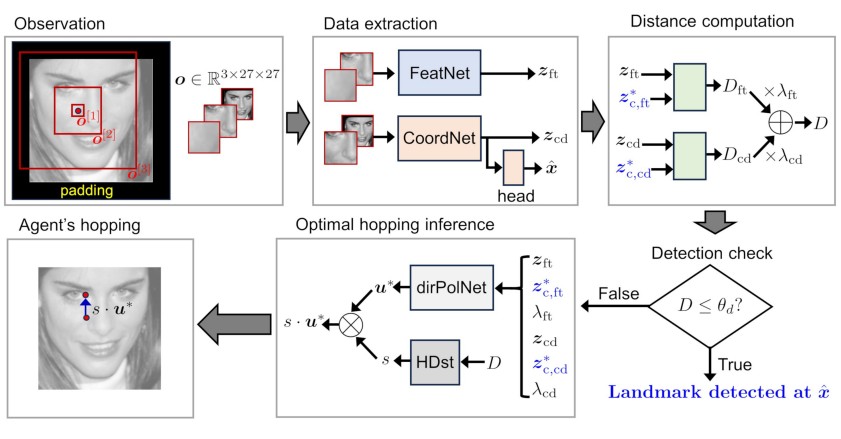

Figure 1: Overview of our algorithm for facial landmark detection.

# 3 METHOD

## 3.1 ALGORITHM OVERVIEW

Our method employs $N_c$ agents, each assigned to search a specific landmark ($c \in \boldsymbol{Cl}$; $|\boldsymbol{Cl}| = N_c$) on a given image of size $C \times H \times W$. In total, the agents simultaneously search all $N_c$ landmarks, with each agent exclusively responsible for one landmark. Their coordinates $\boldsymbol{x} = (x_h, x_w)$ correspond to multiple fixation points, which are normalized to the image size ($x_h, x_w \in [-1, 1]$). We also use normalized coordinate for landmarks. Agents have access to:

1. Prior knowledge of landmarks in both the coordinate and latent feature spaces.

2. Multiscale local observations $\boldsymbol{o} \in \mathbb{R}^{3C \times a \times a}$, with $a \ll H, W$.

Agents **do not** know their absolute positions; positions are inferred from observations $\boldsymbol{o}$. No inter-agent communication occurs. Fig. 1 illustrates the framework at timestep $t$.

**Observation by agents.** To let each agent understand how the local visual structure relates to its spatial context, we consider three patches ($o^{[1]}$: $C \times a \times a$, $o^{[2]}$: $C \times s_1 a \times s_1 a$, and $o^{[3]}$: $C \times s_2 a \times s_2 a$; $1 < s_1 < s_2$) centered at its current location $\boldsymbol{x}$. The $o^{[2]}$ and $o^{[3]}$ patches are resized to $C \times a \times a$ and concatenated with the $o^{[1]}$ patch to construct the observation $\boldsymbol{o} \in \mathbb{R}^{3C \times a \times a}$. Throughout this study, we set $a = 27$, $s_1 = 4$, and $s_2 = 10$, respectively. When the patches exceeds the original image boundaries, Constant padding is applied when patches exceed image boundaries.

**Data extraction from local observation.** Observation $\boldsymbol{o}$ by an agent located at $\boldsymbol{x}$ is processed to extract the latent feature and coordinate of the local view in its spatial context using the following models.

- FeatNet : $\mathbb{R}^{2C \times a \times a} \to \mathbb{R}^{d_{\mathrm{ft}}}$. FeatNet projects current local observation $\boldsymbol{o}$ by an agent at $\boldsymbol{x}$ to an embedding $\boldsymbol{z}_{\mathrm{ft}} \in \mathbb{R}^{d_{\mathrm{ft}}}$. This network is trained using proximity-weighted contrastive learning, which projects observation $\boldsymbol{o}$ to an embedding similar to spatially proximal landmarks. Tensor $\boldsymbol{o}_{0:2C,:,:}$ is provided as input. $\boldsymbol{z}_{\mathrm{ft}} = \mathrm{FeatNet}(\boldsymbol{o}_{0:2C,:,:})$. We omit $\boldsymbol{o}_{2C:3C,:,:}$ due to its minimal contribution. Unless otherwise stated, we fix $d_{\mathrm{ft}} = 128$.

- CoordNet : $\mathbb{R}^{2C \times a \times a} \to \mathbb{R}^{2+d_{\mathrm{cd}}}$, which infers the agent's current absolute coordinate. CoordNet function infers the agent's current coordinate $\hat{\boldsymbol{x}}$ from the local observation, which is also trained using proximity-weighted contrastive learning. Simultaneously, CoordNet projects the observation to a $d_{\mathrm{cd}}$-dimensional vector $\boldsymbol{z}_{\mathrm{cd}}(\equiv \hat{\boldsymbol{x}})$. We omit $\boldsymbol{o}_{0:C,:,:}$ due to its minimal contribution. $[\hat{\boldsymbol{x}}, \boldsymbol{z}_{\mathrm{cd}}] = \mathrm{CoordNet}(\boldsymbol{o}_{C:3C,:,:})$. We fix $d_{\mathrm{cd}} = 128$.

- RelCoordNet : $\mathbb{R}^{2(C+1) \times a \times a} \to \mathbb{R}^{2+d_{\mathrm{cd}}}$. This function infers the agent's current relative coordinate ($\Delta \hat{\boldsymbol{x}}$; $\Delta \hat{x}_h, \Delta \hat{x}_w \in [-2, 2]$) with reference to a given coordinate $\boldsymbol{x}^0$. We used RelCoordNet instead of CoordNet for landmarks with high positional variability.

---

**Algorithm 1** Generative model for prior knowledge.

---

**Input**: Training dataset $\mathcal{T}$ of $N_\mathcal{T}$ samples.
**Output**: Generative model parameters $\boldsymbol{\mu}_{\boldsymbol{z}_{\text{ft}}|c}, \boldsymbol{\sigma}^2_{\boldsymbol{z}_{\text{ft}}|c}, \boldsymbol{\mu}_{\boldsymbol{z}_{\text{cd}}|c}, \boldsymbol{\sigma}^2_{\boldsymbol{z}_{\text{cd}}|c}$

1: $\boldsymbol{K}_{\text{ft}} \leftarrow$ zero tensor of shape $(N_\mathcal{T}, N_c, d_{\text{ft}})$
2: $\boldsymbol{K}_{\text{cd}} \leftarrow$ zero tensor of shape $(N_\mathcal{T}, N_c, d_{\text{cd}})$
3: **for** each sample $y_i$ in $\mathcal{T}$ **do**
4:    **for** each landmark $c_j$ in $\boldsymbol{C}$ **do**
5:       Compute $\boldsymbol{o}$ for landmark $c_j$ in sample $y_i$
6:       $K_{\text{ft}}[i, j] \leftarrow \text{FeatNet}(\boldsymbol{o}_{0:2C,:,:})$
7:       $K_{\text{cd}}[i, j] \leftarrow \text{CoordNet}(\boldsymbol{o}_{C:3C,:,:})$
8:    **end for**
9: **end for**
10: $\boldsymbol{\mu}_{\boldsymbol{z}_{(\cdot)}|c_j} \leftarrow K_{(\cdot)}.\text{mean}(\dim = 0)$ for $(\cdot) \in \{\text{ft}, \text{cd}\}$
11: $\boldsymbol{\sigma}^2_{\boldsymbol{z}_{(\cdot)}|c_j} \leftarrow K_{(\cdot)}.\text{var}(\dim = 0)$ for $(\cdot) \in \{\text{ft}, \text{cd}\}$

---

**Computation of deviation from knowledge.** For current observation $\boldsymbol{o}$, the latent feature and coordinate embeddings ($\boldsymbol{z}_{\text{ft}}$ and $\boldsymbol{z}_{\text{cd}}$) are each compared to their respective preferred embeddings (prior knowledge $\boldsymbol{z}^*_{c,\text{ft}} \in \mathbb{R}^{d_{\text{ft}}}$ and $\boldsymbol{z}^*_{c,\text{cd}} \in \mathbb{R}^{d_{\text{cd}}}$ for landmarks in class $c$) to compute the distance $D$.

$$D = \lambda_{\text{ft}} D_{\text{ft}} + \lambda_{\text{cd}} D_{\text{cd}}, \quad D_{(\cdot)} = ||z_{(\cdot)} - z^*_{c,(\cdot)}||_2^2, \quad \text{where} \quad (\cdot) \in \{\text{ft}, \text{cd}\}. \tag{1}$$

The distance $D$ is a weighted combination of distances from each embedding space, using balance parameters ($\lambda_{\text{ft}}$ and $\lambda_{\text{cd}}$) such that $\lambda_{\text{ft}} + \lambda_{\text{cd}} = 1$. If the distance $D$ is lower than a preset threshold $\theta_d$, the agent has successfully arrived at the landmark, outputting its current coordinate ($\hat{\boldsymbol{x}}$ if CoordNet was used, and $\boldsymbol{x}^0 + \Delta\hat{\boldsymbol{x}}$ if RelCoordNet was used). Otherwise, the agent continues searching.

**Hopping policy.** The agent infers optimal hopping direction and distance using PolNet that is factorized into two sub-functions (dirPolNet and HDst): PolNet $= \text{HDst} \cdot \text{dirPolNet}$.

- dirPolNet : $\mathbb{R}^{2d_{\text{ft}}+2d_{\text{cd}}+2} \rightarrow \mathbb{R}^8$. dirPolNet is a categorical classifier that infers the optimal hopping direction $\boldsymbol{u}^* \in \boldsymbol{U}$. We defined the direction space $\boldsymbol{U}$ as follows: $\boldsymbol{U} = \{(u_1, u_2) \,|\, (u_1 \in \boldsymbol{U}_0 \vee u_2 \in \boldsymbol{U}_0) \wedge (u_1, u_2 \neq 0)\}$, where $\boldsymbol{U}_0 = \{-1, 0, 1\}$. Therefore, $|\boldsymbol{U}| = 8$. This model takes $\boldsymbol{z}_{\text{ft}}/\boldsymbol{z}^*_{c,\text{ft}}/\lambda_{\text{ft}}$ and $\boldsymbol{z}_{\text{cd}}/\boldsymbol{z}^*_{c,\text{cd}}/\lambda_{\text{cd}}$ for the current observation $\boldsymbol{o}$ as input.

- HDst : $\mathbb{R} \rightarrow \mathbb{Z}_{>0}$. This function determines the hopping distance $s \in [1, s_{\max}]$ as a function of the distance $D$.

## 3.2 PRIOR KNOWLEDGE MODELING

We built generative models for prior knowledge of landmarks projected to two independent spaces (latent feature space $\mathbb{R}^{d_{\text{ft}}}$ and coordinate space $\mathbb{R}^{d_{\text{cd}}}$).

$$p(\boldsymbol{z}_{\text{ft}}, c) = p(\boldsymbol{z}_{\text{ft}}|c)\, p(c) \quad \text{for latent features}, \quad p(\boldsymbol{z}_{\text{cd}}, c) = p(\boldsymbol{z}_{\text{cd}}|c)\, p(c) \quad \text{for coordinate},$$

where $c$ denotes a landmark class. We modeled the class-conditional probability distribution function $p(\boldsymbol{z}|c)$ using a Gaussian function:

$$p(\boldsymbol{z}|c) = \mathcal{N}(\boldsymbol{\mu}_{z|c}, \Sigma_{z|c}) \approx \mathcal{N}(\boldsymbol{\mu}_{z|c}, \boldsymbol{\sigma}^2_{z|c}\boldsymbol{I}), \tag{2}$$

where we apply a diagonal approximation to the covariance matrix $\Sigma_{\boldsymbol{z}|c}$ for computational simplicity. The parameters $\boldsymbol{\mu}_{z|c}$ and $\boldsymbol{\sigma}^2_{z|c}$ were separately computed for each of the two embeddings (to $\boldsymbol{z}_{\text{ft}}$ and $\boldsymbol{z}_{\text{cd}}$) over all landmarks of the same class in a given training dataset. This procedure is detailed in **Algorithm** 1.

## 3.3 NETWORK MODELS

**FeatNet.** This model (2C9(6C9)-9C16-16C32-32C64-FC256-FC128-L2Norm for gray-scale (RGB) images) projects the local observation $\boldsymbol{o}_{0:2C,:,:}$ by an agent at $\boldsymbol{x}$ to the feature space $\mathbb{R}^{d_{\text{ft}}}$.

Inspired by supervised contrastive learning (Khosla et al., 2020), we trained FeatNet using a novel proximity-weighted contrastive learning algorithm that locates the embedding $z_{\text{ft}}$ for a given observation close to landmarks of spatial proximity. To this end, all landmarks within a $o^{[2]}$ patch of $C \times s_1 a \times s_2 a$ size centered at given $x$ are considered as positive landmarks while the others as negative ones. Furthermore, we defined the degree of positiveness for the positive landmarks based on their distances from the coordinate $x$.

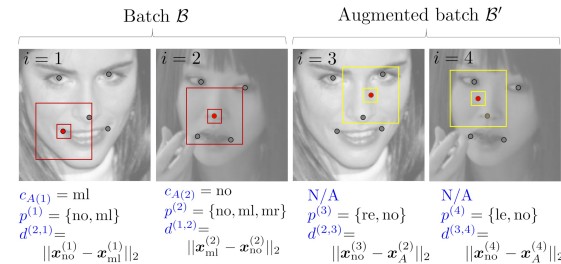

For proximity-weighted contrastive learning, the $i$th sample in a given batch $\mathcal{B}$ includes a single anchor at $x_A^{(i)}$, which is placed on a landmark $c_{A(i)}$ (at $x_{c_{A(i)}}^{(i)}$) that is randomly sampled from total $N_c$ landmarks ($x_A^{(i)} = x_{c_{A(i)}}^{(i)}$). We define an augmented batch $\mathcal{B}'$ of the same samples (and sequence) as $\mathcal{B}$ but with a random anchor for each sample, and thus unnecessarily $x_A^{(i)} = x_{c_{A(i)}}^{(i)}$. The key is the use of a proximity-weighted contrastive loss (PWConLoss) for $\mathcal{B}$ and $\mathcal{B}'$.

Figure 2: Example of sample augmentation for proximity-weighted contrastive learning. Right eye, left eye, nose, mouth left, and mouth right are denoted by re, le, no, ml, and mr, respectively.

$$\mathcal{L}^{\text{PWS}} = \frac{-1}{|\mathcal{B}|} \sum_{i \in \mathcal{B}} \left[ \underbrace{\frac{1}{N^{(i)}} \sum_{j \in \mathcal{B} \setminus \{i\}} w^{(i,j)} \mathbb{1}_{\left\{ c_{A(i)} \in p^{(j)} \right\}} l^{(i,j)}}_{\text{between } c_{A(i)} \text{ and } c_{A(j)} \ (j \neq i)} + \underbrace{\frac{1}{N'^{(i)}} \sum_{j \in \mathcal{B}'} w^{(i,j)} \mathbb{1}_{\left\{ c_{A(i)} \in p^{(j)} \right\}} l^{(i,j)}}_{\text{between } c_{A(i)} \text{ and random observations } o} \right],$$

$$N^{(i)} = \sum_{j \in \mathcal{B} \setminus \{i\}} \mathbb{1}_{\left\{ c_{A(i)} \in p^{(j)} \right\}}, \ N^{'(i)} = \sum_{j \in \mathcal{B}'} \mathbb{1}_{\left\{ c_{A(i)} \in p^{(j)} \right\}}, \ l^{(i,j)} = \log \frac{\exp(z_A^{(i)} \cdot z_A^{(j)}/\tau)}{\sum\limits_{k \in \mathcal{B}_s \setminus \{i\}} \exp(z_A^{(i)} \cdot z_A^{(k)}/\tau)}, \quad (3)$$

where $p^{(j)} = \left\{ c \in Cl | x_c^{(j)} \in o^{[2]} \text{ for } x_A^{(j)} \right\}$. The weight $w^{(i,j)}$ is given by

$$w^{(i,j)} = 1 + \exp\left( -0.025 d^{(i,j)} \right), \quad (4)$$

where $d^{(i,j)}$ denotes the distance between $x_{c_{A(i)}}^{(j)}$ and $x_A^{(j)}$. Note that $x_{c_{A(i)}}^{(j)}$ means the coordinate of the anchor landmark type of the $i$th sample $c_{A(i)}$ on the $j$th sample. For $j \in \mathcal{B} \setminus \{i\}$, the equality $x_A^{(j)} = x_{c_{A(j)}}^{(j)}$ holds. In Eq. 3, $\mathcal{B}_s = \text{concat}(\mathcal{B}, \mathcal{B}')$, and $z_{\text{c,ft}}^{(i)}$ and $\tau$ denote the embedding of the anchor in the $i$th sample and temperature, respectively. The weight $w^{(i,j)}$ in Eq. 4 is constrained to $(1, 2]$. The constant $0.025$ is chosen so that $w^{(i,j)} = 1.5$ when $d^{(i,j)} = 27$, matching the height/width of the smallest patch $o^{[1]}$ of size $C \times 27 \times 27$. An example of sample augmentation for proximity-weighted contrastive learning is shown in Fig. 2.

**CoordNet/RelCoordNet.** CoordNet projects the local observation $o_{C:3C,:,:}$ by an agent at $x$ to the coordinate space $\mathbb{R}^{d_{\text{cd}}}$. It consists of four convolutional layers and one linear layer: 2C9(6C9)-9C16-16C32-32C64-FC128-L2Norm for gray-scale (RGB) images. We deploy an additional head for coordinate regression, Linear$(128 \rightarrow 2)$ + Tanh, which infers the normalized coordinate $\hat{x}$ ($\hat{x}_h, \hat{x}_h \in [-1, 1]$) for the agent at $x$. Similar to FeatNet, the main network (except the head) is trained using proximity-weighted contrastive learning using the PWConLoss in Eq. 3 with a weight function $w^{(i,j)} = 2 - 9.26 \cdot 10^{-3} d^{(i,j)}$ instead of Eq. 4. We adopt a linear weight since the weight needs to linearly scale with the distance, clipped to $[1, 2]$. We used the constant $9.26 \cdot 10^{-3}$ to set $w^{(i,j)} = 1$ at $d^{(i,j)} = 108$, corresponding to the width/height of the smallest patch $o^{[2]}$ of size $C \times 108 \times 108$ in CoordNet/RelCoordNet. The anchor for each samples in batch $\mathcal{B}$ is placed on a random coordinate on the sample unlike FeatNet (for which the anchor is on a landmark only), and thus, the embedding $z_{\text{cd}}$ for a given observation becomes similar to other observations of spatial proximity. The additional head is trained using a mean squared error loss function (MSELoss).

RelCoordNet has the same architecture as CoordNet except the first convolutional layer and head for coordinate regression: 4C9-9C16-16C32-32C64-FC128–L2Norm-FC2-2Tanh for gray-scale im-

---

**Algorithm 2** Delayed decision algorithm.

---

**Input**: $\hat{\mathbf{\Lambda}}$, $D_{\text{ft}}$, $D_{\text{cd}}$, $\theta_{\text{d}}$, $\lambda_{\text{ft}}$, $\hat{x}$
**Output**: Updated SHT and $\hat{x}_c$
1: $D_{\min} \leftarrow$ MAX; $\hat{x}_c \leftarrow$ NULL
2: **for** $i = 0$ to $N_\lambda - 1$ **do**
3: $\quad D_{\text{tmp}} \leftarrow \hat{\mathbf{\Lambda}}[i]D_{\text{ft}} + (1 - \hat{\mathbf{\Lambda}}[i])D_{\text{cd}}$
4: $\quad$ **if** $\text{SHT}[i,0] > D_{\text{tmp}}$ **then**
5: $\quad\quad \text{SHT}[i,0] \leftarrow D_{\text{tmp}}$; $\text{SHT}[i,1] \leftarrow \hat{x}$
6: $\quad$ **end if**
7: **end for**
8: $i \leftarrow 0$
9: **while** $\hat{\mathbf{\Lambda}}[i] \geq \lambda_{\text{ft}}$ **do**
10: $\quad$ **if** $\text{SHT}[i,0] \leq \theta_{\text{d}}$ and $\text{SHT}[i,0] \leq D_{\min}$ **then**
11: $\quad\quad \hat{x}_c \leftarrow \hat{x}$; $D_{\min} \leftarrow \text{SHT}[i,0]$
12: $\quad$ **end if**
13: $\quad i \leftarrow i + 1$
14: **end while**

---

ages. RelCoordNet takes the reference coordinate $x^0$ (represented using $x_w^0 \mathbf{1}_{1 \times a \times a}$ and $x_h^0 \mathbf{1}_{1 \times a \times a}$; $x_w^0, x_h^0 \in [-1, 1]$) as its input alongside the local observation $o_{C:3C,:,:}$, so that the input consists of $2C + 2$ channels. Instead of Tanh, 2Tanh is applied because of the range of relative coordinate $\Delta\hat{x}$ ($\Delta\hat{x}_h, \Delta\hat{x}_w \in [-2, 2]$). This model is also trained using proximity-weighted contrastive learning using PWConLoss in Eq. 3 with random anchors for the samples in batch $\mathcal{B}$ as for CoordNet. The head is trained using MSELoss.

**PolNet.** PolNet infers the optimal hopping direction $u^*$ and distance $s \in [1, s_{\max}]$ for the current local observation $o$ using its sub-functions, dirPolNet and HDst, respectively.

$$\text{PolNet} = \text{HDst}(D) \cdot \text{dirPolNet}(I),$$
$$I = \text{concat}(z_{\text{ft}}, z_{\text{c,ft}}^*, \lambda_{\text{ft}}, z_{\text{cd}}, z_{\text{c,cd}}^*, \lambda_{\text{cd}}), \quad (5)$$
$$\text{HDst}(D) = \min\left(\left(\lceil D/\Delta_D \rceil\right), s_{\max}\right),$$

where $\Delta_D$ denotes a unit step for uniform quantization of distance $D$ in Eq. 1. dirPolNet (FC512-FC256-FC8-Softmax) infers the optimal hopping direction $u^* \in U$, which is trained using supervised learning on a dataset $\mathcal{T}$. We define $U = \{(u_1, u_2) \mid (u_1 \in U_0 \vee u_2 \in U_0) \wedge (u_1, u_2 \neq 0)\}$, where $U_0 = \{-1, 0, 1\}$. That is, $|U| = 8$. For a given image, a pair of coordinate $x$ and landmark $c \in Cl$ are randomly sampled. Similar to the habitual network (Cushman & Morris, 2015), each sample $y_i = (I_i, \hat{u}_i)$ in $\mathcal{T}$ consists of (1) input $I_i$ (in Eq. 5) for the local observation $o$ at the random coordinate $x$ and (2) $\hat{u}_i = \arg\min A_{u \in U} D$ for the coordinate $x$ and landmark $c$.

### 3.4 HYPERPARAMETER SETTING

**Detection threshold.** Detection threshold $\theta_d$ is a primary hyperparameter that determines the detection accuracy and speed, which are measured in NME and duration required for detection ($T_d$), respectively. A lower $\theta_d$ generally yields a lower NME but a larger $T_d$. To balance this trade-off, threshold $\theta_d$ is initially set to its minimum ($\theta_{d,\min}$) and is monotonously increased by $\Delta\theta_d$ at each timestep if detection fails.

**Balance parameters.** Balance parameters $\lambda_{\text{ft}}$ and $\lambda_{\text{cd}}(= 1 - \lambda_{\text{ft}})$ in Eq. 1 govern the complementary contributions of the distances from distinct representation spaces ($D_{\text{ft}}$ and $D_{\text{cd}}$) to the total distance $D$. A higher $\lambda_{\text{ft}}$ generally yields a lower NME but a larger $T_d$. We initially set $\lambda_{\text{ft}}$ is initially set to its maximum($\lambda_{\text{ft,max}}$) and monotonically decrease by $\Delta\lambda_{\text{ft}}$ once every two timesteps if detection fails.

### 3.5 EFFICIENCY ENHANCEMENTS

**Delayed decision algorithm.** Critical detection inefficiency with varying hyperparameters ($\theta_d$ and $\lambda_{\text{ft}}$) arises when previously visited locations with previous hyperparameter values yield successful detection with the current $\theta_d$. Revisiting such locations and repeatedly recalculating the distance $D$

leads to redundant computation. To mitigate this issue, we introduce a delayed decision algorithm for the following case.

$$\theta_{\mathrm{d}}[t'] < D[t'] = \sum_{i \in \{\mathrm{ft,cd}\}} \lambda_i[t']D_i[t'] \le \theta_{\mathrm{d}}[t] \text{ for } t' < t.$$

We define a balance parameter set $\Lambda = \{\lambda[t]|\forall t \in [0, t_{\max}]\}$ ($|\Lambda| = N_\lambda$) and a corresponding tuple $\hat{\Lambda}$ of $\lambda(\in \Lambda)$ sorted in descending order. This delayed decision algorithm is based on a $N_\lambda \times 2$ search history table (SHT). **Algorithm** 2 explains SHT organization and delayed decision based on the SHT.

**Two-stage detection.** To reduce the detection duration $T_{\mathrm{d}}$, we introduce a two-stage detection strategy for all landmarks. The first stage detects coarse coordinates of landmarks using the higher detection threshold $\theta_{\mathrm{d}}^{(1)}$ and lower balance parameter $\boldsymbol{\lambda}_{\mathrm{ft}}^{(1)}$, and the second stage refines the coordinate using the lower threshold $\theta_{\mathrm{d}}^{(2)}(< \theta_{\mathrm{d}}^{(1)})$ and higher balance parameter $\boldsymbol{\lambda}_{\mathrm{ft}}^{(2)}(> \boldsymbol{\lambda}_{\mathrm{ft}}^{(1)})$. In each stage, the hyperparameters change following the rule explained in the previous section.

**Cascaded detection.** Facial landmarks can be grouped based on their spatial proximity in the latent feature space $\mathbb{R}^{d_{\mathrm{ft}}}$. We choose a single core-landmark for each group and considered the others in the same group as sub-landmarks. Agents responsible for detecting landmarks in the same group often exhibit overlapping trajectories in the initial detection phase, leading to redundant computation. To address this, we

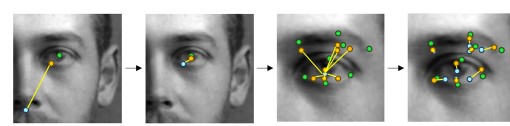

Core-landmark two-stage detection    Sub-landmark two-stage detection

Figure 3: Example of cascaded detection.

propose a cascaded detection strategy. In this strategy, a single agent first detects the core-landmark using FeatNet and CoordNet for local observation $\boldsymbol{o}$ at each timestep $t$. This core-landmark serves as a reference. Subsequently, multiple agents simultaneously search the sub-landmarks with reference to the core-landmark. This approach mitigates redundant agent movements during the early detection steps, thereby reducing redundant computation and processing time. Fig. 3 shows an example of cascaded detection for a Right-eye group.

**Per-stage timeout.** To prevent unbounded roaming of agents, we impose a per-stage timeout (30 steps). If the distance $D$ does not fall below the detection threshold $\theta_d$ within the period, the agent terminates hopping and its position is finalized by the delayed-decision algorithm.

## 4 EXPERIMENTAL RESULTS

We used the COFW (Burgos-Artizzu et al., 2013) and 300W (Sagonas et al., 2016) datasets as a proof of concept. COFW comprises 1,345 training and 507 test images (gray-scale), each annotated with 29 landmarks. COFW with frequent occlusions is well-suited for evaluating our method's ability to leverage prior knowledge of landmarks. 300W comprises 3,148 training and 689 test images (RGB), each annotated with 68 landmarks. This dataset exhibits a wide range of variations in pose and lighting conditions.

### 4.1 IMPLEMENTATION DETAILS

Each image is cropped to include the full head, resized to $256 \times 256$, then randomly rescaled ($\pm 5\%$) and horizontally flipped ($50\%$). For cascaded detection, the landmarks in each dataset are grouped based in their spatial proximity as follows.
**COFW**: `left_eye` (left pupil), `right_eye` (right pupil), and `others` (nose tip)
**300W**: `left_eye` (left inner canthus), `right_eye` (right inner canthus), `mouth` (Cupid's bow), `nose` (nose tip), `jaw_line` (none).
The landmarks in parentheses indicate core landmarks. We used different FeatNets with different sets of parameters (but the same CoordNet and RelCoordNet) for each group. For the Others group in COFW, RelCoordNet was used for detecting the sub-landmarks. Note that we used the means $\boldsymbol{\mu}_{\boldsymbol{z}_{\mathrm{ft}|c}}$ and $\boldsymbol{\mu}_{\boldsymbol{z}_{\mathrm{cd}|c}}$ as prior knowledge $\boldsymbol{z}_{\mathrm{c,ft}}^*$ and $\boldsymbol{z}_{\mathrm{c,ft}}^*$, respectively, unless otherwise stated. The models were trained using the Pytorch framework (Paszke et al., 2019) on a GPU workstation (RTX A6000;

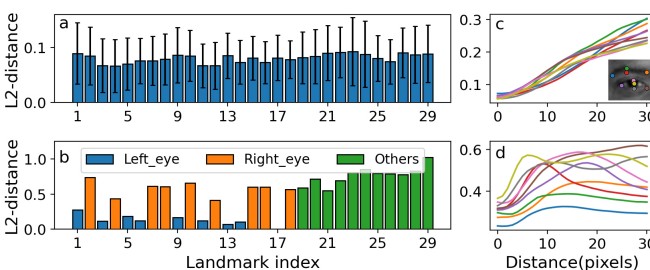

Figure 4: Performance of FeatNet embeddings. (**a**) L2 distances between embeddings of landmarks within the same class. (**b**) L2 distance between a left-pupil embedding and embeddings of other landmarks. (**c**) L2 distances as a function of spatial distance from a given landmark. (**d**) Comparison with a supervised contrastive learning baseline.

Figure 5: L2 distances between coordinate embeddings and landmarks at varying spatial distances from the landmarks in the inset for (**a**) CoordNet and (**b**) RelCoordNet.

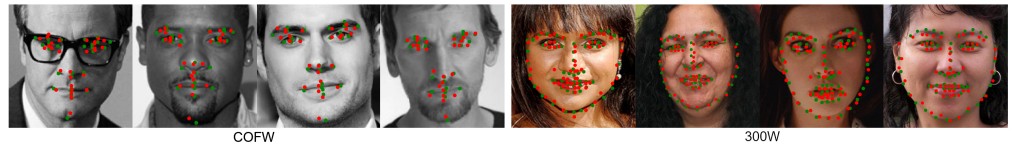

Figure 6: Detected landmarks (red circles) and ground-truth annotations (green circles) on sample images from COFW and 300W.

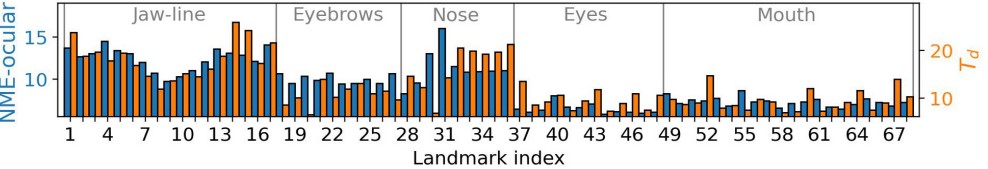

Figure 7: Average detection accuracy (NME) and detection duration for each landmark on 300W.

Xeon Gold CPU 2.9GHz; 256 GB DRAM). Landmark detection experiments were conducted on both the GPU workstation and a desktop equipped with an i5 CPU (2.5GHz) and 32 GB DRAM. The hyperparameters were optimized using Optuna (Akiba et al., 2019). All hyperparameters are summarized in Appendix.

### 4.2 PERFORMANCE OF NETWORK MODELS

**FeatNet.** We analyzed a fully trained FeatNet whose learning curve is shown in Appendix. Fig. 4**a** shows L2 distance between landmarks in the same classes in COFW, identifying successful landmark clustering. Fig. 4**b** shows L2 distance between a left pupil (Class 17) and the others on the same image. This identifies the separation of landmark clusters based on their spatial proximity. We analyzed the capability of FeatNet to encode observations $o$ at random coordinates $x$ as $z_{\text{ft}}$ based on their distances from landmarks. Fig. 4**c** highlights (1) a gradual increase in L2 distance with distance between the observation $o$ and landmark and (2) marginal variability in L2 distance at aero distance upon different landmarks. As a counterpart, we used the same network trained with supervised contrastive learning, where spatial proximity was addressed in binary form, employing the same loss as in Eq. 3 but without the proximity weight $w^{(i,j)}$. The performance of this counterpart is shown in Fig. 4**d**.

**CoordNet/RelCoordNet.** Fully trained CoordNet and RelCoordNet (whose learning curves are shown in Appendix) successfully infer the coordinate of the current observation as $z_{\text{cd}}$-dimensional embeddings. Fig. 5 shows the L2 distance between the coordinate embedding $z_{\text{cd}}$ and several land-

Table 1: Comparison of our method with SoTA approaches on COFW and 300W datasets. The NME value in parentheses for 300W excludes the `jaw_line` landmarks.

| Method | COFW | | 300W | # Params (M) | FLOPs | Duration $T_d$ |
| | NME-ocular | NME-pupil | NME-ocular | | | |
|---|---|---|---|---|---|---|
| LAB (Wu et al., 2018) | 3.92 | 5.58 | 3.49 | 25.1 | 18.9G | - |
| AWing (Wang et al., 2019) | - | 4.94 | 3.07 | 24.2 | 26.8G | - |
| AVS (Qian et al., 2019) | - | 4.43 | 3.86 | 28.3 | 2.40G | - |
| LaplaceKL (Robinson et al., 2019) | - | - | 3.91 | 2.20 | 12.5G | - |
| DAG (Li et al., 2020) | - | - | 4.27 | 21.0 | - | - |
| HRNet (Wang et al., 2020) | 3.45 | - | 3.32 | 9.66 | 4.75G | - |
| PIP (Jin et al., 2021) | - | - | 3.36 | 12.0 | 2.40G | - |
| ADNet (Huang et al., 2021) | - | 4.69 | 2.93 | 13.4 | 17.0G | - |
| SDFL (Lin et al., 2021) | 3.63 | - | - | - | 5.17G | - |
| HIH (Lan et al., 2021) | 3.21 | 4.63 | 3.09 | 22.7 | 17.2G | - |
| SLPT (Xia et al., 2022) | 3.32 | 4.63 | 3.17 | 13.2 | 6.12G | - |
| STARLoss (Zhou et al., 2023) | - | 4.62 | 2.87 | 13.4 | - | - |
| D-ViT (Dang et al., 2025) | - | 4.13 | 2.85 | 67.3 | 21.8G | - |
| PoPos (Xiang et al., 2025) | - | 3.80 | 3.28 | 9.70 | 1.20G | - |
| **Ours on COFW/300W** | **8.28** $\pm$**0.08** | **11.96** $\pm$**0.12** | **9.36 (8.33)** $\pm$**0.05** | **0.577** | **21.1**$\pm$**0.8M /** **29.1**$\pm$**1.1M** | **10.42**$\pm$**0.36 /** **12.77**$\pm$**0.52** |

Table 2: FPS on different processors.

| | CPU i5 2.5GHz | CPU Xeon 2.9GHz | GPU A6000 |
|---|---|---|---|
| COFW | 4.19$\pm$0.11 | 3.00$\pm$0.14 | 19.73$\pm$1.15 |
| 300W | 1.29$\pm$0.19 | 1.25$\pm$0.15 | 5.21$\pm$0.68 |

marks (in the inset) with spatial distance. Compared with Fig. 4, CoordNet/RelCoordnet can infer the coordinate of distal observations with higher precision (lower variability).

### 4.3 DETECTION PRECISION AND EFFICIENCY

The detected landmarks on several samples in COFW and 300W are shown in Fig. 6, demonstrating successful landmark detection using our algorithm. Nevertheless, our method has higher NME than regression-based SoTA techniques as listed in Table 1. This is largely due to the fact that our approach rely on not supervised learning but prior knowledge of landmarks' features. For instance, although our method well detects the jaw-line landmarks in 300W samples (Fig. 6), NME for these landmarks is large due to their deviation from the semi-automatically annotated jaw-line landmarks on 300W samples. Fig. 7 plots the average NME and detection duration $T_d$ for each landmark on 300W sample, identifying high NME for the `jaw_line` landmarks.

However, our method demonstrates significantly low computational complexity.
**Space complexity**: Total 577k parameters (FeatNet/CoordNet/PolNet with 123k/58k/396k parameters).
**Time complexity**: Total 21.1 MFLOPs for a 10.42 $T_d$ on COFW and 29.1 MFLOPs for a 12.77 $T_d$ on 300W. Relative to the lightweight PoPos (Xiang et al., 2025) model with 9.70M parameters, our method reduces space complexity by $16.8\times$ and time complexity by $41.1\times$. Our method with extremely low complexity runs at $4.19$ (COFW) and $1.29$ (300W) frames per second (FPS) on a desktop with an i5-13400 CPU (Table 2).

### 4.4 ABLATION STUDY

**Balance parameter.** The balance parameter $\lambda_{ft}$ is an important hyperparameter that governs both detection accuracy and duration, and its value is scheduled over time. We analyzed the impact of $\lambda_{ft}$ on detection accuracy and duration by varying its value over the range $[0, 1]$ while keeping it fixed during each detection run. As shown in Fig. 8, $\boldsymbol{\lambda}_{ft}$ exhibits a clear trade-off between accuracy and duration, highlighting the need for parameter scheduling to achieve optimal performance.

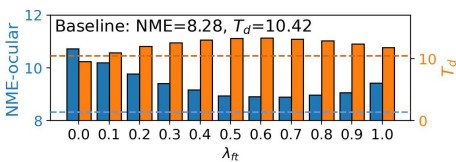

Figure 8: Relationship between detection accuracy and detection duration for different fixed values of $\lambda_{ft}$ on COFW.

Table 3: Impact of techniques on detection performance.

| | Delayed decision | Two-stage detection | Cascaded detection | NME | $T_d$ | FLOPs |
|---|---|---|---|---|---|---|
| Baseline | ✓ | ✓ | ✓ | 8.28 | 10.42 | 21.1M |
| Case 1 | ✗ | ✓ | ✓ | 10.35 | 14.56 | 29.5M |
| Case 2 | ✓ | ✗ | ✓ | 8.52 | 7.81 | 15.8M |
| Case 3 | ✓ | ✓ | ✗ | 9.46 | 20.97 | 42.4M |
| Case 4 | ✗ | ✗ | ✓ | 10.46 | 9.25 | 18.7M |
| Case 5 | ✗ | ✓ | ✗ | 12.50 | 30.32 | 61.4M |
| Case 6 | ✓ | ✗ | ✗ | 9.53 | 16.20 | 32.8M |
| Case 7 | ✗ | ✗ | ✗ | 12.55 | 33.65 | 68.1M |

Table 4: Detection performance on COFW with different prior knowledge models.

| | Baseline | Sampling | Gaussian mixture | From samples |
|---|---|---|---|---|
| NME | 8.28 | 8.92 | 5.99 | 3.88 |
| $T_d$ | 10.42 | 20.35 | 13.41 | 10.39 |

**Performance enhancement techniques.** To enhance detection performance, we employ three techniques: delayed decision, two-stage detection, and cascaded detection. We analyze the impact of each technique on detection accuracy and duration. The corresponding results are presented in Table 3.

**Parameter scheduling and robustness to sample degradations.** The results are shown in Appendix.

## 5 LIMITATIONS AND FUTURE WORK

In our proof-of-concept, each landmark's prior was modeled as a diagonal Gaussian with a single global mean $\mu_{z_{(\cdot)}}$. This likely over-simplifies landmark variability across samples, resulting in lower detection accuracy than state-of-the art. We replaced this with a Gaussian mixture model comprising five components, with means initialized from $K$-means clustering ($K = 5$). Selecting the best component per sample improved accuracy on COFW by approximately $28\%$ as in Table 4. As an upper-bound analysis, using per-sample landmark features as priors yields accuracy close to state of the art (From samples in Table 4). These results indicate that (i) accuracy is primarily limited by the knowledge model and (ii) the other sub-modules (FeatNet, CoordNet, PolNet) function as intended. Our results show a strong correlation between prior knowledge and detection accuracy, suggesting that building a more expressive and robust prior is a key direction for future work.

## 6 CONCLUSION

We proposed a lightweight, agent-based framework for facial landmark detection that leverages prior knowledge and local observations without relying on strong supervision. Each agent infers its location independently using embeddings from dual spaces—feature and coordinate—guided by class-conditional generative models. To train robust embeddings, we introduced proximity-weighted contrastive learning, and we further improved efficiency with a multi-stage detection strategy and delayed decision mechanism to reduce redundant computation. While our method shows slightly higher NME than SoTA approaches, it achieves exceptional efficiency by reducing space complexity by $16.8\times$ and time complexity by $41.1\times$ compared to the SoTA lightweight model, making it ideal for embedded applications. This work demonstrates that prior knowledge-guided agent-based detection is a practical and scalable alternative for efficient landmark localization.

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
