# OpenReview forum: "Efficient Facial Landmark Detection via Prior Knowledge-Guided Agents"
_ICLR.cc/2026/Conference — Submitted to ICLR 2026_

### Official Review · Reviewer_MzRE · 2025-10-31

**Soundness:** 2
**Presentation:** 2
**Contribution:** 2
**Rating:** 4
**Confidence:** 3

**Summary:**

This paper presents a novel, agent-based framework for facial landmark detection that departs from conventional global regression approaches. Each agent is assigned to a specific landmark and navigates the image using only local observations and prior knowledge modeled in dual (feature and coordinate) embedding spaces. The method employs several strategies for efficiency, including a proximity-weighted contrastive loss, a multi-stage detection scheme, and a delayed decision algorithm. While the method's accuracy, measured by NME, is substantially lower than state-of-the-art models, it achieves a dramatic reduction in computational and spatial complexity, making a case for its use in resource-constrained, real-time applications.

**Strengths:**

1. The core paradigm is innovative and thought-provoking. Reformulating landmark detection as a distributed navigation task for independent agents is a non-trivial and intellectually interesting contribution. It opens up a new and potentially fruitful research direction beyond dense prediction with large, monolithic models.
2. The efficiency gains are compelling and quantitatively significant. The paper's most convincing argument is its extreme lightweight nature. The reported >16x and >41x reductions in space and time complexity compared to a modern lightweight baseline are hard, undeniable metrics that strongly support its potential for embedded deployment.

**Weaknesses:**

Prohibitive degradation in accuracy severely limits practical utility. The compromise on precision is arguably too great. The NME on 300W (9.36) is approximately three times higher than current SOTA methods (~2.8-3.3). This level of inaccuracy renders the method unsuitable for most real-world applications requiring precision (e.g., high-fidelity face recognition, medical analysis), significantly diminishing the value of its efficiency.

**Questions:**

1. Fundamental Trade-off: Is the significant accuracy loss an inherent limitation of the agent-based paradigm itself, or can it be substantially mitigated with more sophisticated network designs, training strategies, or prior knowledge modeling? What is your estimate of the method's ultimate accuracy ceiling?
2. Generalization of Priors: Your prior knowledge is modeled as class-conditional Gaussians derived from the training set. Does this mean the method will fundamentally fail on faces with atypical morphology (e.g., extreme expressions, facial anomalies) not well-represented in the training data? This poses a serious limitation to its generalizability.
3. Convergence and Failure Modes: Does the hopping policy guarantee convergence? Please provide an analysis of failure cases, such as agents getting stuck in loops, oscillating between points, or failing to locate occluded landmarks. The reliability of the search process is a critical concern that is not adequately addressed.

---

> ### Author Response · Authors · 2025-11-20
> **Response to Reviewer**
>
> Dear Reviewer, we thank you for your careful review. Below we address the concerns (weaknesses and questions) you raised. All revisions in the manuscript and appendix are highlighted in blue. An editing error in Figure 5 of the original manuscript has been corrected in the revision. We apologize for the oversight.
>
> $\textbf{W1/Q1. limited accuracy}$
>
> Our submission is intended as a first, deliberately simple instantiation of a novel knowledge-based framework for landmark search. We fully acknowledge that the current version underperforms mature baselines. Still, we believe it demonstrates the feasibility of the framework and opens a path toward highly efficient landmark search. To that end, we added targeted ablations to diagnose failure modes and outline concrete accuracy improvements.
>
> $\textbf{Need for a stronger knowledge model.}$ In our proof-of-concept, each landmark’s prior in latent space was modeled as a diagonal Gaussian with a single global mean, which likely over-simplifies landmark variability across samples. During the rebuttal period, we replaced this with a Gaussian mixture model comprising five components, with means initialized from $K$-means clustering ($K=5$). Selecting the best component per sample improved accuracy on COFW by approximately $28\\%$. As an upper-bound analysis, using per-sample landmark features as priors yields accuracy close to state of the art. These results, reported in Sec. 5 (Limitations and Future Work) in the revised manuscript, indicate that (i) accuracy is primarily limited by the knowledge model and (ii) the other sub-modules (FeatNet, CoordNet, PolNet) function as intended.
>
> The manuscript has been updated as follows: Sec. 5 (Limitations and Future Work) explicitly identifies the knowledge model as the primary bottleneck and reports the results mentioned above.
>
> $\textbf{Q2. Generalization of Priors}$
>
> We view prior knowledge as a stereotype, which is useful but inherently brittle to exceptions. To improve robustness, we instantiate complementary priors in two distinct spaces: feature space (FeatNet) and coordinate space (CoordNet). Even under extreme expressions, facial landmark geometry is largely preserved, and for facial anomalies landmarks can be inferred via relative similarity to the prior. No model generalizes perfectly to all edge cases. As noted in $\textbf{W1/Q1}$, the observed accuracy limits arise from our deliberately simple knowledge model rather than from the framework itself.
>
> $\textbf{Q3. Convergence and Failure Modes:}$
>
> Our hopping policy does not guarantee convergence. If the distance $D$ remains above the threshold $\theta_d$, the agent continues to roam. To prevent unbounded roaming, we impose a per-stage timeout (e.g., 30 steps). If $D$ does not fall below the threshold within the period, the agent terminates hopping and its position is finalized by the delayed-decision algorithm using SHT. A representative convergence-failure case under occlusion is analyzed in Fig. 6 of the revised Appendix. We have also added brief explanation of per-stage timeout on Page 7 in the revised manuscript.

---

### Official Review · Reviewer_GYKT · 2025-10-31

**Soundness:** 3
**Presentation:** 3
**Contribution:** 3
**Rating:** 6
**Confidence:** 4

**Summary:**

The paper proposes an **agent-based framework** for facial landmark detection where **each agent localizes a single landmark** using only **local multi-scale observations** and **prior knowledge** modeled as class-conditional Gaussians in **feature** and **coordinate** embedding spaces. Agents iteratively “hop” using a lightweight policy (dirPolNet + distance-quantized step size) until a deviation criterion from priors is satisfied. Training leverages a **proximity-weighted contrastive loss** to encode spatial proximity, and runtime efficiency is boosted via a **two-stage** and **cascaded** detection strategy (core landmark → sub-landmarks with RelCoordNet) plus a **delayed decision** mechanism to avoid redundant recomputation. Experiments on **COFW** and **300W** show substantially lower compute/params (≈ **577k** params; **21–29 MFLOPs**) and CPU FPS up to **4.19** (COFW) / **1.29** (300W), at the cost of higher NME versus SOTA.

**Strengths:**

* **Efficiency first:** ~**577k** parameters and **21–29 MFLOPs** with concrete CPU FPS; strong engineering story for edge inference.

* **Clear mechanism:** Dual priors in feature/coordinate spaces with a principled deviation criterion and a small policy network.

* **Training insight:** **Proximity-weighted** contrastive objective aligns embeddings with spatial closeness, improving robustness to occlusion/noise.

* **System tricks that matter:** Two-stage refinement, cascaded (core→sub) detection, and delayed decision to prune redundant compute.

**Weaknesses:**

* **Accuracy trade-off:** NME notably higher than SOTA (esp. 300W, aggravated on jawline); causes and bounds could be probed more.

* **Stats & variance:** No CIs or multi-seed std; several deltas vs. strong baselines might be within noise without variance reporting.

* **Limited stress testing:** Few controlled perturbations (Gaussian blur, JPEG, motion blur, occlusion masks) to validate the “local-observation + priors” claim under adverse conditions.

* **Ablation scope:** Good focus on λ_ft schedule; less on **policy design** (dirPolNet architecture/targets), **threshold schedule**, and **choice of Gaussian priors vs. richer models**.

* **Interpretability:** Nice qualitative plots, but no **quantitative** measure that agents consistently specialize by region/frequency or that cascades reduce token/patch evaluations by X%.

* **Missing References:** Several works should be included. Particularly, heatmap-based landmark detection approaches. E.g.,

- Robinson, J. P., Li, Y., Zhang, N., Fu, Y., & Tulyakov, S. (2019). Laplace landmark localization. In Proceedings of the IEEE/CVF international conference on computer vision (pp. 10103-10112).

Though several others.

**Questions:**

1. **Statistical robustness.** Please report **mean±std over ≥3 seeds** (or bootstrap CIs) for NME and duration on COFW/300W; do the efficiency–accuracy trade-offs hold?

    16871_Efficient_Facial_Landmar

2. **Controlled degradations.** How does NME change under blur (σ grid), JPEG (Q grid), motion blur, and synthetic occlusion masks? Does proximity-weighted contrastive training confer robustness?

    16871_Efficient_Facial_Landmar

3. **Policy ablations.** What is the impact of: (i) different direction sets (e.g., 16-way), (ii) continuous step sizes vs. quantized HDst, and (iii) removing λ_ft/θ_d scheduling?

    16871_Efficient_Facial_Landmar

4. **Priors.** Why diagonal Gaussians? Would **mixture** priors or **low-rank covariances** help multi-modal landmark appearance/pose? Any cost increase?

    16871_Efficient_Facial_Landmar

5. **Cascade efficiency.** Can you quantify wall-clock and FLOP savings from the cascaded detection (core→sub) relative to parallel independent agents?

    16871_Efficient_Facial_Landmar

6. **Generalization.** Any results on non-frontal or in-the-wild video frames (e.g., low-light, motion) to demonstrate edge robustness?

    16871_Efficient_Facial_Landmar

---

> ### Author Response · Authors · 2025-11-20
> **Response to Reviewer**
>
> Dear Reviewer, we thank you for your careful review. Below we address the concerns (weaknesses and questions) you raised. All revisions in the manuscript and appendix are highlighted in blue. An editing error in Figure 5 of the original manuscript has been corrected in the revision. We apologize for the oversight.
>
> $\textbf{W1. Accuracy trade-off}$
>
> In our proof-of-concept, each landmark’s prior in latent space was modeled as a diagonal Gaussian with a single global mean, which likely over-simplifies landmark variability across samples. During the rebuttal period, we replaced this with a Gaussian mixture model comprising five components, with means initialized from $K$-means clustering ($K=5$). Selecting the best component per sample improved accuracy on COFW by approximately $28\\%$. As an upper-bound analysis, using per-sample landmark features as priors yields accuracy close to state of the art. These results, reported in Sec. 5 (Limitations and Future Work) in the revised manuscript, indicate that (i) accuracy is primarily limited by the knowledge model and (ii) the other sub-modules (FeatNet, CoordNet, PolNet) function as intended.
>
> The manuscript has been updated as follows: Sec. 5 (Limitations and Future Work) explicitly identifies the knowledge model as the primary bottleneck and reports the results mentioned above.
>
> $\textbf{W2/Q1. Stats and variance}$
>
> We agree with you. Accordingly, we have conducted additional experiments with three seeds to evaluate the statistics of performance. The results are summarized in Table 1 in the manuscript, which highlight marginal variability in performance.
>
> $\textbf{W3/Q2. Limited stress testing}$
>
> Following your suggestion, we have validated the robustness of our method to sample degradations for ten cases in total. The results are summarized in Table 4 in Appendix. While blurring with $\sigma=3$ causes the largest NME loss, it does not exceed $7.5\\%$. This may justify the robustness of our method.
>
> $\textbf{W4/Q3/Q4. Ablation scope}$
>
> $\textbf{Policy ablations:}$ We have not yet validated PolNet with a larger hopping space ($>8$). We will keep you updated if they become available before the rebuttal deadline. Previously, we attempted to model PolNet with real-value outputs instead of using quantized HDst. But, its performance was not comparable to quantized HDst. Regarding hyperparameter scheduling, we observed a $\times 2.69$ increase in latency without $\theta_\text{d}$ scheduling and a $\times 3.27$ increase without both $\theta_\text{d}$ and $\lambda_\text{ft}$ scheduling. The results are summarized in Table 3 in Appendix.
>
> $\textbf{Priors:}$ We chose a diagonal Gaussian as a knowledge model to minimize the time and space complexity. As responded to $\textbf{W1}$, a Gaussian mixture model certainly improves the detection accuracy, implying that a Gaussian with a single mean (irrespective of full consideration of covariances) over-simplifies landmark variability across samples. However, this causes difficulty in choosing the optimal Gaussian component for a given sample. The additional cost of Gaussian mixture is marginal.
>
> $\textbf{W5/Q5. Interpretability (cascade efficiency)}$
>
> You can find the significant FLOPs saving from Cascaded detection by comparing Baseline with Case 3 in Table 3 in the revised manuscript. Without Cascaded detection, the number of FLOPs and the detection latency increase by $\times 2$.
>
> $\textbf{W6. Missing References}$
>
> We thank you for the suggestion. Yes, we have added two references including Robinson, J.P. et al. and address their performance in Table 1 in the revised manuscript.
>
> $\textbf{Q6. Generalization}$
>
> Following your suggestion, we have evaluated our method on WFLW (Wider Facial Landmarks in-the-Wild). The evaluated performance is listed in Table 10 in Appendix in comparison with several state of the art.

---

### Official Review · Reviewer_2GrR · 2025-11-01

**Soundness:** 3
**Presentation:** 2
**Contribution:** 2
**Rating:** 4
**Confidence:** 3

**Summary:**

This paper introduces a novel, agent-based framework for facial landmark detection, prioritizing computational efficiency and model compactness for edge devices. The framework involves independent agents localizing landmarks, guided by learned prior knowledge modeled as Gaussian distributions in both feature and coordinate spaces. While explicitly trading accuracy for efficiency, the experiments show significantly higher NME than existing state-of-the-art methods.

**Strengths:**

1. This paper novelly proposes dual feature and coordinate priors and a proximity-weighted contrastive loss (PWConLoss).
2. The framework shows exceptional model compactness and low computational cost, achieving 16.8× lower space complexity and 41.1× lower time complexity than the best lightweight baseline.
3. The concept of using completely independent agents reframes the landmark detection problem from a single large inference to a collection of lightweight, parallelizable searches.

**Weaknesses:**

1. The claim of "real-time CPU inference" is misleading. According to Table 2, the highest FPS on CPU is only 4.19, far from the common "real-time" definition (>20 FPS).
2. Severe accuracy sacrifice. According to Table 1, the proposed framework's NME is almost three times larger than that of the state-of-the-art baseline.
3. The experiments lack comparison with other real-time methods, e.g., MediaPipe Face Mesh.

**Questions:**

Please refer to the weakness part.
1. Please clarify the "real-time CPU inference" claim.
2. Is there any idea or plan to improve the accuracy?
3. Could you include some comparison with other real-time methods?

---

> ### Author Response · Authors · 2025-11-20
> **Response to Reviewer**
>
> Dear Reviewer, we thank you for your careful review. Below we address the concerns (weaknesses and questions) you raised. All revisions in the manuscript and appendix are highlighted in blue. An editing error in Figure 5 of the original manuscript has been corrected in the revision. We apologize for the oversight.
>
> $\textbf{W1/Q1. misleading term}$
>
> We fully agree with you and apologize for the carelessness. Accordingly, we have removed the term "real-time" in the revision.
>
> $\textbf{W2/Q2. Low accuracy and plans for improvement}$
>
> Our submission is intended as a first, deliberately simple instantiation of a novel knowledge-based framework for landmark search. We fully acknowledge that the current version underperforms mature baselines. Still, we believe it demonstrates the feasibility of the framework and opens a path toward highly efficient landmark search. To that end, we added targeted ablations to diagnose failure modes and outline concrete accuracy improvements.
>
> $\textbf{Need for a stronger knowledge model.}$ In our proof-of-concept, each landmark’s prior in latent space was modeled as a diagonal Gaussian with a single global mean, which likely over-simplifies landmark variability across samples. During the rebuttal period, we replaced this with a Gaussian mixture model comprising five components, with means initialized from $K$-means clustering ($K=5$). Selecting the best component per sample improved accuracy on COFW by approximately $28\\%$. As an upper-bound analysis, using per-sample landmark features as priors yields accuracy close to state of the art. These results, reported in Sec. 5 (Limitations and Future Work) in the revised manuscript, indicate that (i) accuracy is primarily limited by the knowledge model and (ii) the other sub-modules (FeatNet, CoordNet, PolNet) function as intended.
>
> The manuscript has been updated as follows: Sec. 5 (Limitations and Future Work) explicitly identifies the knowledge model as the primary bottleneck and reports the results mentioned above.
>
> $\textbf{W3/Q3. Missing comparison with real-time methods}$
>
> The open-source MediaPipe achieves $>$20 FPS on the same CPU for COFW and 300W. However, we cannot report its detection accuracy on these datasets because the model was trained on a different dataset with different landmark definitions. Moreover, the dataset used to train MediaPipe is not publicly available, so we cannot train our model on the same data. Therefore, we cannot quantitatively compare our method with MediaPipe for the moment. Instead, we briefly overview MediaPipe in Sec. 2.1 (Regression-based methods) on Page 2 in the revised manuscript.

---

### Official Review · Reviewer_hoGW · 2025-11-01

**Soundness:** 2
**Presentation:** 3
**Contribution:** 2
**Rating:** 4
**Confidence:** 4

**Summary:**

This paper proposes an agent-based framework for facial landmark detection that emphasizes computational efficiency and model compactness over state-of-the-art accuracy. Each landmark is assigned to an independent agent that localizes its target through local observations and learned priors, without inter-agent communication. The prior knowledge is modeled as class-conditional Gaussian distributions in both latent feature and coordinate embedding spaces. A proximity-weighted contrastive loss (PWConLoss) is further introduced to improve the robustness of feature embeddings by weighting positive samples according to their spatial proximity. The framework also employs multi-stage and cascaded detection strategies to enhance efficiency. Experiments on COFW and 300W demonstrate notable reductions in parameters (16.8×) and FLOPs (41.1×) compared with lightweight baselines, highlighting a strong trade-off between accuracy and efficiency.

**Strengths:**

Exceptional Efficiency: The most notable strength lies in the substantial improvement in computational and memory efficiency. Achieving over a 40× reduction in FLOPs and a 16× reduction in parameters compared with a lightweight state-of-the-art model (PoPos) is highly impressive, making the method exceptionally practical for real-world deployment on edge devices.

Clarity and Presentation: The paper is written with excellent clarity. The authors effectively present a complex system with well-structured explanations and thorough mathematical derivations.

**Weaknesses:**

Significant Accuracy Drop: Despite the strong efficiency gains, the accuracy falls far below the competitive range, making the method impractical for accuracy-sensitive applications. Efficiency improvements lose significance when core task performance is not maintained.

Unclear Accuracy–Efficiency Trade-off: The trade-off between accuracy and efficiency is not convincingly analyzed. Comparisons should control for a single variable—e.g., reducing parameters in existing models or increasing those of the proposed one—to fairly assess the balance.

High Complexity and Limited Ablation: The framework integrates multiple heuristic components (three networks, two-stage and cascaded strategies, delayed-decision buffer, and several hyperparameters), making it difficult to analyze, tune, and reproduce. Its performance likely depends heavily on these design choices. Only λft is ablated, with no analysis of other major modules.

Unexplained Parameters: Certain parameters, such as the constant 0.025 in Equation (4), are introduced without justification.

**Questions:**

Please refer to the weakness.

---

> ### Author Response · Authors · 2025-11-20
> **Response to Reviewer**
>
> Dear Reviewer, we thank you for your careful review. Below we address the concerns you raised. All revisions in the manuscript and appendix are highlighted in blue. An editing error in Figure 5 of the original manuscript has been corrected in the revision. We apologize for the oversight.
>
> $\textbf{W1. Significant accuracy drop}$
>
> Our submission is intended as a first, deliberately simple instantiation of a novel knowledge-based framework for landmark search. We fully acknowledge that the current version underperforms mature baselines. Still, we believe it demonstrates the feasibility of the framework and opens a path toward highly efficient landmark search. To that end, we added targeted ablations to diagnose failure modes and outline concrete accuracy improvements.
>
> $\textbf{Need for a stronger knowledge model.}$ In our proof-of-concept, each landmark’s prior in latent space was modeled as a diagonal Gaussian with a single global mean, which likely over-simplifies landmark variability across samples. During the rebuttal period, we replaced this with a Gaussian mixture model comprising five components, with means initialized from $K$-means clustering ($K=5$). Selecting the best component per sample improved accuracy on COFW by approximately $28\\%$. As an upper-bound analysis, using per-sample landmark features as priors yields accuracy close to state of the art. These results, reported in Sec. 5 (Limitations and Future Work) in the revised manuscript, indicate that (i) accuracy is primarily limited by the knowledge model and (ii) the other sub-modules (FeatNet, CoordNet, PolNet) function as intended.
>
> The manuscript has been updated as follows: Sec. 5 (Limitations and Future Work) explicitly identifies the knowledge model as the primary bottleneck and reports the results mentioned above.
>
> $\textbf{W2. Unclear Accuracy–Efficiency Trade-off}$
>
> Increasing method complexity yields only marginal accuracy gains because performance is ultimately bounded by the knowledge model (as noted above). We have not yet compiled accuracy–efficiency trade-off data for existing baselines. We will include these results if they become available before the submission deadline (December 3).
>
> $\textbf{W3. High Complexity and Limited Ablation}$
>
> We agree and have added extensive ablations:
>
> $\bullet$ Effect of removing one or more of: (i) two-stage detection, (ii) cascaded detection, (iii) delayed decision (Table 3 in the revised manuscript).
>
> $\bullet$ Effect of disabling the schedules for $\lambda_{\mathrm{ft}}$ and/or $\theta_{\mathrm{d}}$ (Table 3 in the revised Appendix).
>
> $\bullet$ Robustness to sample degradations (Table 4 in the revised Appendix).
>
>
>
> $\textbf{W4. Unexplained Parameters}$
>
> The exponential weight $w^{(i,j)}$ in Eq. (4) increases feature contrast with spatial distance and is constrained to $(1,2]$. The constant $0.025$ is chosen so that $w^{(i,j)}=1.5$ when $d^{(i,j)}=27$, matching the height/width of the smallest patch $o^{[1]}$ of size $C\times27\times27$.
>
> For CoordNet/RelCoordNet, which encodes spatial distances, we adopt a linear weight, $w^{(i,j)}=a-bd^{(i,j)}$, with positive constants $a,b$, clipped to $[1,2]$. We set $w^{(i,j)}=1$ at $d^{(i,j)}=108$, corresponding to the width/height of the smallest patch $o^{[2]}$ of size $C\times108\times108$ in CoordNet/RelCoordNet.
>
> The revised manuscript now includes this explanation on page 5.

---

### Meta-Review · Area_Chair_tyA7 · 2026-01-06

**Summary:**

- The authors propose an agent-based framework for facial landmark detection, where agents localize each landmark independently by navigating the image using local observations and learned Gaussian priors.
- Reviewers acknowledge the novelty of framing facial landmark detection as an agent-based navigation problem, as well as the high computational efficiency demonstrated on the presented benchmarks, though direct comparisons with state-of-the-art real-time methods are unclear.
- Notably, all reviewers raise serious concerns over the substantial degradation in accuracy w.r.t existing methods. Results from the rebuttal indicate that increasing model capacity and incorporating more complex knowledge models fail to recover competitive performance. Therefore, it appears the current approach lacks a clear path toward closing the performance gap with existing methods, limiting its potential of applicability and suitability for publication at this time.

**Reviewer Concerns:**

- In the revised version, authors added ablation studies to cascaded detection, delayed decision, etc.
- The authors show that generalization to extreme poses, low-light, etc., is preserved.
- A failure case analysis on the convergence of navigation is provided.

Outstanding issues:
- The authors removed claims to "real-time" performances, though they did not address how this affects their claim to efficiency.
- The authors respond to the common concerns of accuracy drop by attributing it to a simpler knowledge model. They indicate that more complex knowledge models, such as a Gaussian mixture model, would lead to improvements, though still not comparable to the performance of existing methods. The accuracy-efficiency trade-off remains unclear. Increasing model complexity does not lead to significant improvement. Therefore, there is not a clear path towards competitive performance.

**Reviewer Scores:**

No reviewers had replied to the rebuttals prior to the cutoff. Due to the mismatched expectation to model performance on landmark detection, reviewers likely would have kept their scores.

---

### Decision · Program_Chairs · 2026-01-26

Reject